# Egg Consumption and 4-Year Change in Cognitive Function in Older Men and Women: The Rancho Bernardo Study

**DOI:** 10.3390/nu16162765

**Published:** 2024-08-19

**Authors:** Donna Kritz-Silverstein, Ricki Bettencourt

**Affiliations:** 1Herbert Wertheim School of Public Health and Longevity Science, University of California San Diego, La Jolla, CA 92093-0725, USA; 2Department of Family Medicine, School of Medicine, University of California San Diego, La Jolla, CA 92093-0725, USA; 3Division of Gastroenterology and Hepatology, School of Medicine, University of California San Diego, La Jolla, CA 92093-0725, USA; rbettencourt@health.ucsd.edu

**Keywords:** cognitive decline, cognitive function, cognitive performance, egg consumption, memory, older men and women

## Abstract

The effect of dietary cholesterol on cognitive function is debatable. While eggs contain high levels of dietary cholesterol, they provide nutrients beneficial for cognitive function. This study examined the effects of egg consumption on change in cognitive function among 890 ambulatory adults (N = 357 men; N = 533 women) aged ≥55 years from the Rancho Bernardo Study who attended clinic visits in 1988–1991 and 1992–1996. Egg intake was obtained in 1988–1991 with a food frequency questionnaire. The Mini-Mental Status Exam (MMSE), Trails B, and category fluency were administered at both visits to assess cognitive performance. Sex-specific multiple regression analyses tested associations of egg intake with changes in cognitive function after adjustment for confounders. The mean time between visits was 4.1 ± 0.5 years; average ages were 70.1 ± 8.4 in men and 71.5 ± 8.8 in women (*p* = 0.0163). More men consumed eggs at higher levels than women; while 14% of men and 16.5% of women reported never eating eggs, 7.0% of men and 3.8% of women reported intakes ≥5/week (*p* = 0.0013). In women, after adjustment for covariates, egg consumption was associated with *less* decline in category fluency (beta = −0.10, *p* = 0.0241). Other associations were nonsignificant in women, and no associations were found in men. Results suggest that egg consumption has a small beneficial effect on semantic memory in women. The lack of decline observed in both sexes suggests that egg consumption does not have detrimental effects and may even have a role in the maintenance of cognitive function.

## 1. Introduction

Rates of memory impairment and Alzheimer’s disease have been increasing [1]. Prevalence estimates for 2024 suggest that in the United States, over 6.9 million people over the age of 65 years have Alzheimer’s disease [1]. The greatest risk factor for memory impairment and Alzheimer’s disease is age [1,2], and given the increasing longevity of people who are alive today, the prevalence of this disease is expected to almost double to about 13.8 million by 2060 [1,3]. Because aging cannot be changed, identifying modifiable behaviors that can affect the risk of cognitive decline is crucial.

According to a review and meta-analysis of seventeen studies, high plasma cholesterol in midlife more than doubles the risk of mild cognitive impairment and Alzheimer’s disease in later years [4]. However, the association of dietary cholesterol with cognitive decline is inconsistent [5,6,7,8,9,10]. Although some cross-sectional studies report that greater intake of foods rich in cholesterol is associated with lower cognitive performance [5,6], other studies report either no effect [7,8,9] or a beneficial effect [10,11]. 

Although eggs contain high levels of cholesterol, unlike meats, they have low levels of fat. They are also rich in protein, amino acids, nutrients such as choline, and carotenoids (e.g., lutein and zeaxanthin), which are needed for cognitive function and protect against cognitive decline [12,13,14,15,16,17]. However, the results of the three prior studies that addressed the longitudinal association of egg intake with cognitive function are inconsistent. For instance, in a representative sample of 3835 men and women from the United States aged 65 years and older, egg consumption was not significantly associated with cognitive function performance two years later [18]. In contrast, among 480 men in Finland aged 42–60 years, egg consumption at baseline was associated with better performance on measures of executive function and verbal fluency when assessed four years later [7]. Finally, in our recently published study using data from 1515 older adults in the Rancho Bernardo cohort with cognitive function evaluated more than 16 years after assessment of egg intake, we found that men who consumed more eggs had better total, short-term, and long-term recall, but no effects were found in women [19]. In a similar dearth of research, only one previous study examined the relation of egg intake with changes in cognitive performance over time [9]. In that study, egg intake was not associated with global cognitive decline in either sex. However, that study was limited to individuals aged 50 to 70 years, and there was only an average of 2.1 years between cognitive assessments [9]. 

This study is aimed at investigating the association of egg intake with a 4-year change in cognitive function in a large sample of community-dwelling older men and women. 

## 2. Materials and Methods

### 2.1. Participants

Data for this study come from members in the Rancho Bernardo Study, a prospective study that enrolled community-dwelling, middle- to upper-middle-class individuals living in a southern California community in 1972–1974. These individuals have been followed with research clinic visits at approximately 4-year intervals. 

Eligible study participants were 1314 older individuals who had egg intake and cognitive performance assessed in a 1988–1991 clinic visit; cognitive function was reassessed at a 1992–1996 clinic visit. After excluding those younger than age 55 (N = 92), missing information on egg consumption in 1988–1991 (N = 359), and reporting having had a stroke (N = 39), there were a total of 890 individuals (357 men and 533 women) remaining for inclusion in this report. 

The Human Research Protections Program of the University of California San Diego approved this study and all clinic visits (IRB #191902); all participants were community-dwelling, ambulatory, and provided written informed consent before participating. 

### 2.2. Cognitive Assessment

A standardized set of cognitive function assessments was first administered to Rancho Bernardo cohort members in 1988–1991. These tests were chosen in conjunction with the Alzheimer’s Disease Research Center at the University of California San Diego specifically because they addressed l different domains of cognitive function that were most likely to be sensitive to aging. Due to time constraints and the desire to avoid subject fatigue, only a subset of these tests was administered at subsequent visits. At both the 1988–1991 and 1992–1996 visits, trained interviewers administered three tests of cognitive function. The Mini-Mental State Examination (MMSE) [20,21] evaluates global cognitive function, including orientation, attention, registration, calculation, language, and recall, with scores ranging from 0–30; higher scores indicate better performance [20,21]. The Trail-Making Test, Part B (Trails B), [22] assesses executive function, including visuomotor tracking, attention, and mental flexibility. Individuals are given a sheet containing scattered numbers and letters, each within its own circle, and asked to alternate connecting numbers to letters in ascending order (e.g., 1 to A, A to 2, 2 to B, B to 3, 3 to C, and so on). Participants are given a maximum of 300 s to complete this task, with scores representing the time (seconds) required to finish; higher scores indicate *poorer* performance [22]. Category fluency, a test of verbal fluency, evaluates semantic memory and executive function [23]. Individuals are given 1 min during which they must name as many animals as they can. Scores reflect the number of animals correctly named after excluding repetitions, variants (i.e., dogs after producing dog), and intrusions (e.g., apple); higher scores are indicative of better performance [23].

### 2.3. Dietary Assessment

Egg consumption was assessed in 1988–91 with the Willet Food Frequency Questionnaire (FFQ) [24]. Participants were given a list of 153 foods and asked to indicate for each one how often it was consumed and the specific portion size usually consumed during the previous year. Response choices for eggs were never, 1–3/month, 1/week, 2–4/week, 5–6/week, 1/day, 2–3/day, 4–5/day, and 6+/day. Because of low frequencies, the highest categories of egg consumption were combined into the category of ≥5/week. Estimates of daily intakes for each nutrient were obtained using the Harvard nutrient database program, which multiplies responses by the nutrient compositions of the corresponding portion sizes of each food (HarvardSSFQ.5/93; Harvard TH Chan School of Public Health, Boston, MA, USA). Total calories and total protein intake per day were obtained in this manner. FFQ estimates were considered unlikely and set to missing if daily caloric intake was under 600 or over 4200 kcal [25]. 

### 2.4. Covariates

A standardized self-administered questionnaire was used to obtain information on age and the number of years of education completed. Participants were also queried about their lifestyle and behaviors, including cigarette smoking status (never/past/current), alcohol consumption ≥ 3 times/week (no/yes), exercise ≥ 3 times/week (no/yes), which were obtained with this self-administered questionnaire. Medication use (including medication for hypertension, diabetes, and high cholesterol) was validated with pills and containers brought to the clinic for that purpose. Medical history was obtained, including physician diagnosis of diabetes (no/yes). Blood pressure (BP) was measured after the participant had been seated quietly for five minutes by a nurse trained in the Hypertension Detection Follow-up Protocol [26]; the mean of two readings for systolic and diastolic BPs was used. Plasma levels of total cholesterol, fasting glucose (FPG), and 2 h postchallenge glucose (PCPG) were measured at a previous (1984–1987) research clinic visit. Briefly, lipids were measured in fasting blood samples in a Lipid Research Clinics laboratory under continuous standardization of the Centers for Disease Control in Atlanta, GA. Fasting plasma total cholesterol level was measured by an enzymatic technique with an ABA-200 biochromatic analyzer (Abbot Laboratories, Irving TX) and a high-performance cholesterol reagent (No. 236691, Boehringer-Mannheim Diagnostics, Indianapolis, IN, USA). Fasting and postchallenge plasma glucose levels were measured with a glucose oxidase method in a diabetes research laboratory.

### 2.5. Statistical Analysis

Education was categorized into ≤high school vs. ≥some college. Current cigarette smoking was categorized into no vs. yes. Participants were considered to have hypertension if they had a systolic blood pressure ≥ 140 mmHg, diastolic blood pressure ≥ 90, or used antihypertensive medications. Diabetes status was defined by WHO criteria [27] as FPG > 126 mg/dL, PCPG > 200 mg/dL, the use of medication for diabetes, or a reported physician diagnosis. 

Means and standard deviations were calculated for all continuous variables, and rates were calculated for all categorical variables. Comparisons of men and women on continuous variables were performed with independent *t*-tests and on categorical variables, including categorical egg intake, with chi-square analysis. Because of significant differences found between men and women on the independent and dependent variables, and on almost all covariates, multivariable analyses were all sex-specific. Comparisons of characteristics by categorical egg consumption (never/1–2 per month/1 per week/2–4 per week/5 or more per week) were performed separately within men and within women using analysis of variance for continuous variables and chi-square analysis or Fisher’s exact test for categorical variables. Change scores for each cognitive function test were calculated for each person as follows:

(1992–1996 cognitive test score − 1988–1991 cognitive test score)/time between visits. Sex-specific linear regression analyses were used to evaluate the relation of egg consumption with cognitive function change scores. For each test, different models successively adjusted for covariates: model 1 adjusted associations for age and education (demographics); model 2 adjusted for model 1 variables + current smoking, alcohol, exercise, and cholesterol (behaviors); and model 3 adjusted for model 2 variables + total calories and protein intake. Other models added adjustments for diabetes and hypertension. 

SAS (version 9,4, SAS Institute, Cary, NC, USA) was used to perform all statistical analyses. Statistical tests were all two-tailed with *p*-values < 0.05 needed for significance. 

## 3. Results

In both sexes, the mean time between visits was 4.1 ± 0.5 years (medians = 4.0 years for women, 3.9 years for men). Average age was 70.1 in men and 71.5 in women (*p* = 0.0163). Comparisons of other characteristics in 1988–1991 (Table 1) indicated that men had greater education (*p* < 0.0001) and calorie intake (*p* < 0.0100), as well as higher rates of alcohol use (*p* < 0.0001), exercise (*p* = 0.0082), and diabetes (*p* = 0.0408), but lower total cholesterol (*p* < 0.0001) than women. There were no differences by sex in rates of smoking, hypertension, and protein intake (*p*’s > 0.10). Compared to women in both 1988–1991 and 1992–1996, men performed significantly better on Trails B (*p* < 0.0001 and *p* = 0.0008, respectively) and category fluency (*p*’s < 0.0001). However, men performed worse than women on the MMSE, a difference that was significant in 1992–1996 (*p* = 0.0400). In both sexes, Trails B and category fluency scores declined over time. 

Comparisons of categorical egg intake by sex (Figure 1) showed that 14.0% of men and 16.5% of women reported never eating eggs in the previous year, while 7% of men and 3.8% of women reported consuming five or more eggs per week. There were significant sex differences in egg intake, with higher rates of men consuming 2–4 eggs/week and ≥5 eggs/week than women and higher rates of women not consuming eggs or consuming only 1–3 eggs/month than men (*p* = 0.0013). 

Sex-specific comparisons of demographics, behaviors, diet, and health by categorical egg intake (Table 2) indicated that for both sexes, those who consumed five or more eggs per day had the highest levels of calorie and protein intake. Additionally, within both sexes, those who ate 2–4 eggs/week had the lowest cholesterol levels. Men who reported never consuming eggs in the past year had the lowest rates of drinking alcohol ≥ 3 times/week. No other differences by categorical egg intake were observed.

Successive regression models examined covariate-adjusted associations of egg intake with a 4-year change in cognitive performance separately in men and women (Table 3). In women, after adjustment for demographic variables, egg consumption was associated with significantly less decline in category fluency. Specifically, each category increase in egg consumption was related to a 0.10 less decrease in category fluency (*p* = 0.0100). Thus, women in the highest category of egg intake would have a half-point less decline in category fluency score over 4 years compared to women who never consumed eggs. Although small, this association was still significant after further adjustment for behaviors, cholesterol, and calorie and protein intake (Table 3), as well as after additional adjustment for diabetes and hypertension. There were no associations of egg intake with a 4-year change in the MMSE or Trails B for women, and there were no associations of egg intake with a 4-year change in any of the cognitive test scores in men before and after covariate adjustment (*p*’s > 0.10).

## 4. Discussion

### 4.1. Study Outcomes

Given the increasing life span of individuals and the large financial, personal, and societal burdens imposed by the increasing prevalence of cognitive impairment, it is imperative that factors that could prevent or delay cognitive decline with age are identified. In this large sample of older adults from a well-characterized cohort, with cognitive performance assessed twice over 4 years, we found that among women, greater egg intake was related to significantly less decline in category fluency, a test of semantic memory and executive function. Although this difference was small, it persisted even after controlling for variables known to affect cognitive function, such as age, education, current cigarette smoking status, alcohol use, exercise, plasma cholesterol, and total calorie and protein intake, as well as the presence of diabetes and hypertension. Other associations were nonsignificant in women, and no associations of egg intake with changes in cognitive function were found in men. Thus, egg consumption was not associated with a decline in performance and possibly contributed to the maintenance of cognitive function over time. This is only the second study that we know of that investigates the association of egg consumption with cognitive change and the first to examine this issue in a US cohort of older individuals using sex-specific analyses.

The absence of a relation of egg intake with a decline in cognitive performance among men in the present study is in agreement with the lack of association with either incident dementia or Alzheimer’s disease over 21.9 years in a study among 2497 men aged 42–60 in Finland [7]. Likewise, the present study’s results agree with those from a previous analysis of data from this cohort in which we reported that egg intake was unrelated to the risk of impaired cognitive performance as assessed over 16 years later [19]. However, in that analysis, cognitive performance was only assessed at one time point, whereas this study specifically examined egg consumption in relation to changes in the same cognitive function measures repeated at two time points.

To the best of our knowledge, only one previous study specifically examined the relation of egg intake with changes over time in cognitive function [9]. In that study, 2514 Chinese men and women aged 50–70 years were given measures of global cognitive function, auditory verbal learning, and other tests of recall and followed for 2.1 years. No association was found between egg intake and the risk of decline on any cognitive function test. However, the use of different cognitive function measures, a shorter duration of follow-up (2.1 years), and a lack of sex-specific analyses along with the use of a cohort that was much younger and less educated (median age = 59 years, median education = 9 years), and from a different country than participants in the Rancho Bernardo cohort, create challenges when trying to compare the findings of the two reports. 

In the current study, numerous sex differences were observed. For instance, women consumed less alcohol and had lower rates of diabetes, which should protect against cognitive decline, but they also had less education and exercised less, which would have had the opposite effect. Although both sexes have shown declines in Trails B and category fluency scores over time, at both visits, men outperformed women on these measures, whereas women had better performance on the MMSE. However, women declined less than men in category fluency over four years (differences = 1.2 vs. 1.9, respectively). Men consumed more calories than women, and a greater proportion of men reported eating eggs at higher levels. However, egg consumption was associated with *less* 4-year decline in verbal fluency only in women—a sex difference that cannot be explained. 

### 4.2. Biologic Plausibility

Several lines of evidence suggest that it is biologically plausible that egg consumption has a role in the maintenance or prevention of a decline in cognitive performance. Eggs have high levels of protein, amino acids, and cholesterol, which may help preserve neuronal structure and function within the brain [28]. Additionally, eggs are one of the richest sources of choline, which is a precursor to the neurotransmitter, acetylcholine, which is necessary for cognitive function [29]. Cross-sectional studies have shown that those with greater choline intake and those with higher plasma concentrations of choline had better scores on several measures of cognitive function [13,30,31]. Additionally, a 12-week clinical trial of older adults (aged 50–85 years) with age-associated memory impairment found that those receiving dietary supplementation with citicoline, a naturally occurring mononucleotide that contains choline, performed better on cognitive tasks [14]. 

Eggs are rich in carotenoids, including lutein and zeaxanthin, which have been implicated in the maintenance of cognitive performance [17,32]. In samples of older adults, including centenarians, those with greater serum and brain concentrations of lutein and zeaxanthin had significantly higher scores on tests of executive function, language, and episodic memory [17,33]. Although the mechanism is not completely understood, it has been postulated that these carotenoids may be beneficial as a result of their anti-inflammatory or anti-oxidant effect on the brain [17,33]. Additionally, clinical trials among individuals with Alzheimer’s disease reported that those given lutein and zeaxanthin supplementation had less progression of disease [34,35]. Healthy older adults who received supplementation also showed improvement in attention and other measures of cognitive performance [36,37].

### 4.3. Limitations and Strengths

This study is not without its potential limitations. The generalizability of this study’s results may be limited due to the homogeneity of the Rancho Bernardo Study participants, who are predominantly White, highly educated, and can afford medical care. However, this homogeneity is advantageous as it means that differences due to culture, education, and the ability to afford medical care are less likely to have confounded performance on cognitive function tests. Furthermore, previous analyses comparing data from this cohort with data from national, representative samples of older adults show that while participants in the Rancho Bernardo Study have a somewhat lower rate of obesity [38], they have similar rates of cigarette smoking [39], alcohol intake, blood pressure, total cholesterol, diabetes, and impaired glucose tolerance [40,41,42,43,44]. 

Several variables used in this study, including the assessment of egg consumption, were based on self-reports, which can be subject to recall bias and other inaccuracies. However, self-reports of behaviors have been validated for subsets of this cohort using medical records or by clinical or laboratory assessments. For example, self-reported alcohol use in this cohort correlated directly with dietitian-assessed alcohol intake as well as indirectly via its positive associations with plasma aspartate aminotransferase and HDL [40,45]. Reported cigarette use was validated indirectly by its strong association with impaired pulmonary function [46]. Reported exercise was indirectly corroborated by its negative relation with pulse rate and direct relation with HDL [47]. 

This analysis involved numerous statistical tests, increasing the possibility that results were due to chance. However, all comparisons were planned a priori, the association of egg intake with less decline in cognitive performance was consistently observed for category fluency, and none of the other tests showed an increase in decline. The 4-year duration of follow-up was relatively short, albeit twice as long as the only other study examining the association of egg intake with cognitive function change. Given the older age of this cohort, the follow-up duration was unlikely to have limited our ability to detect differences in cognitive function.

The American Heart Association guidelines suggesting the limitation of egg intake were first introduced in 1968, and the widespread adoption of these guidelines by the late 1980s may have limited egg intake and, consequently, our ability to detect associations. However, a relatively wide range of egg intake was still observed. As with all studies of older individuals, we cannot exclude the possibility of participation bias due to selective attrition of individuals with the most memory impairment, nor can we exclude the possibility of survival bias, where the oldest participants died before they had an opportunity to attend the follow-up. However, participation bias and survival bias would have yielded conservative estimates of the results. The results of this study do not suggest benefits for choline, lutein, and zeaxanthin, which may play a role in cognitive health in particular and not eggs in general. Lastly, because brain imaging studies were not performed, we are unable to relate our observed results to physical brain changes. 

Numerous other strengths of this study should be considered, such as the large sample size of older adults, assessment of several domains of cognitive performance with standardized tests, and performance of sex-specific analyses. Additionally, statin medication use was unlikely to have biased the results obtained here as the assessment of egg intake in 1988–1991 was before the more pervasive prescribing of statins. 

## 5. Conclusions

This study found that in women, those who consumed more eggs per week had less decline over 4 years in semantic memory and executive function. Egg intake was unrelated to a decline in performance on other cognitive tests in women and men. The lack of cognitive decline with egg consumption is reassuring and suggests that despite having high levels of dietary cholesterol, eggs do not have a detrimental effect and may even have a role in the maintenance of cognitive function over time. Eggs are readily available and a low-cost option to obtain protein and nutrients linked to health in other studies. Future longitudinal studies of large samples of older individuals should include sex-specific analyses to either confirm or refute the results of this study. Additionally, studies using brain imaging could be used to determine whether the lack of change in cognitive performance over time with egg consumption is consistent with an absence of changes observed in the brain. 

## Figures and Tables

**Figure 1 nutrients-16-02765-f001:**
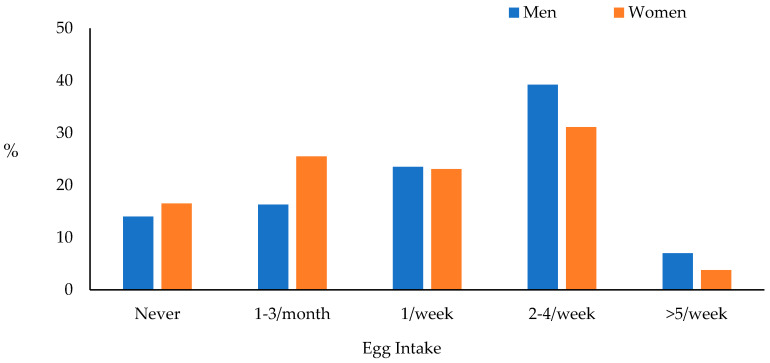
Comparisons of categorical egg intake by sex; Rancho Bernardo, CA 1988–1991, results of chi-square analysis, *p* = 0.0013. N per category in men vs. women, respectively: Never = 50, 88; 1–3/month = 58, 136; 1/week = 84, 123; 2–4/week = 140, 166; ≥5/week = 25, 20.

**Table 1 nutrients-16-02765-t001:** Comparisons between men and women on demographics, behaviors, diet, health, and cognitive function.

Characteristic 1988–1991	Men (N = 357)	Women (N = 533)	*p*-*Value* ^d^
Demographics			
Age (mean ± sd)	70.1 ± 8.4	71.5 ± 8.8	**0.0163**
College education (%)	82.1	63.6	**<0.0001**
Behaviors			
Current smoking (%)	9.0	8.8	0.9074
Alcohol ≥ 3x/week (%)	58.3	44.5	**<0.0001**
Exercise ≥ 3x/week (%)	79.4	71.6	**0.0082**
Health			
Cholesterol (mg/dl, mean ± sd)	214.4 ± 39.3	227.9 ± 38.3	**<0.0001**
Cholesterol lowering meds (%)	6.0	3.4	0.0836
Diabetes ^a^ (%)	15.5	10.8	**0.0408**
Hypertension ^b^ (%)	67.0	62.3	0.1538
Diet-related			
Calories/day (mean ± sd)	1816.3 ± 506.4	1696.7 ± 560.7	**0.0010**
Protein/day (mean ± sd)	74.3 ± 22.9	72.3 ± 24.3	0.2281
Cognitive function 1988–1991			
Mini-Mental State (mean ± sd)	27.5 ± 1.7	27.7 ± 1.4	0.0796
Trails B ^c^ (mean ± sd)	105.9 ± 45.9	121.6 ± 55.9	**<0.0001**
Category Fluency (mean ± sd)	20.2 ± 4.8	18.2 ± 4.7	**<0.0001**
Cognitive function 1992–1996			
Mini-Mental State (mean ± sd)	27.8 ± 2.5	28.1 ± 1.8	**0.0400**
Trails B ^c^ (mean ± sd)	124.9 ± 56.2	138.8 ± 64.1	**0.0008**
Category Fluency (mean ± sd)	18.3 ± 5.0	17.0 ± 4.7	**<0.0001**

^a^ Diabetes = fasting plasma glucose > 126, post = challenge plasma glucose > 200, reported use of anti-diabetic medication at 1984–87 clinic visit, or self-reported diabetes or anti-diabetic medication use at 1988–91. ^b^ Hypertension = systolic BP ≥ 140, diastolic BP ≥ 90, or anti-hypertensive medication use. ^c^ For Trails B, higher scores indicate poorer performance. ^d^ Bolded *p-values* are statistically significant.

**Table 2 nutrients-16-02765-t002:** Comparisons of demographics, behaviors, diet, and health by egg intake in men and women; Rancho Bernardo, CA 1988–1991.

Characteristic	Egg Consumption 1988–1991					
	Never	1–3/Month	1/Week	2–4/Week	5+/Week	*p*-Value ^c^
MEN	N = 50	N = 58	N = 84	N = 140	N = 25	
Demographics						
Age (mean ± sd)	70.5	70.6	68.9	70.3	70.9	0.6665
College education (%)	78.0	89.7	84.3	76.6	95.6	0.0680
Behaviors						
Current smoking (%)	2.0	15.8	10.7	7.1	12.0	0.1174
Alcohol ≥ 3x/week (%)	44.0	72.4	58.3	56.4	64.0	**0.0499**
Exercise ≥ 3x/week (%)	88.0	82.8	75.9	77.7	76.0	0.4469
Health						
Cholesterol (mean ± sd)	222.7	224.1	213.4	206.8	221.3	**0.0243**
Cholesterol medication(%)	13.6	6.1	2.9	4.2	9.5	0.1381
Diabetes ^a^ (%)	6.0	3.5	15.5	10.0	12.0	0.1546
Hypertension ^b^ (%)	38.0	31.0	31.0	34.3	28.0	0.8761
Diet						
Calories/day (mean ± sd)	1724.8	1573.8	1791.4	1897.8	2189.2	**<0.0001**
Protein/day (mean ± sd)	69.9	61.0	74.2	78.3	91.7	**<0.0001**
	**Never**	**1–3/Month**	**1/Week**	**2–4/Week**	**5+/Week**	** *p* ** **-value ^c^**
**WOMEN**	**N = 88**	**N = 136**	**N = 123**	**N = 166**	**N = 20**	
Demographics						
Age (mean ± sd)	70.4	72.0	71.3	71.9	71.2	0.7273
College education (%)	66.3	67.2	62.6	60.6	60.0	0.7713
Behaviors						
Current smoking (%)	7.0	8.2	9.8	8.6	15.0	0.8178
Alcohol ≥ 3x/week (%)	36.4	45.6	46.3	46.4	45.0	0.5860
Exercise ≥ 3x/week (%)	72.7	74.1	68.8	70.5	75.0	0.8897
Health						
Cholesterol (mean ± sd)	236.8	231.3	225.3	222.0	232.0	**0.0439**
Cholesterol medication (%)	8.3	2.3	1.8	3.2	0	0.1589
Diabetes ^a^ (%)	3.4	2.9	8.9	4.8	5.0	0.2115
Hypertension ^b^ (%)	28.4	44.1	42.2	35.2	30.0	0.1075
Diet						
Calories/day (mean ± sd)	1511.4	1490.0	1714.8	1906.8	2056.2	**<0.0001**
Protein/day (mean ± sd)	67.7	63.5	70.9	80.8	90.9	**<0.0001**

^a^ Diabetes = fasting plasma glucose > 126, post = challenge plasma glucose > 200, reported use of anti-diabetic medication at 1984–87 clinic visit, or self-reported diabetes or anti-diabetic medication use at 1988–91. ^b^ Hypertension = systolic BP ≥ 140, diastolic BP ≥ 90, or anti-hypertensive medication use. ^c^ Bolded *p-values* are statistically significant.

**Table 3 nutrients-16-02765-t003:** Sex-specific covariate-adjusted associations of egg intake (1988–1991) with 4-year cognitive function change ^a^; Rancho Bernardo, CA 1988–1991 to 1992–1996.

	Men		Women	
	Βeta	*p*-Value	Βeta	*p*-Value
4-Year Change ^a^				
MMSE				
Model 1	0.00	0.9757	−0.00	0.9677
Model 2	0.01	0.8255	−0.00	0.9366
Model 3	0.00	0.9083	−0.00	0.9691
Trails B ^b^				
Model 1	−0.19	0.6879	−0.00	0.9998
Model 2	−0.25	0.6060	−0.02	0.9719
Model 3	−0.12	0.8110	−0.33	0.5717
Category Fluency				
Model 1	0.01	0.7801	−0.10	**0.0100**
Model 2	0.01	0.8613	−0.10	**0.0108**
Model 3	0.01	0.8680	−0.10	**0.0241**

Results of multiple regression analysis. Model 1—adjusted for age and education. Model 2—adjusted for model 1 variables and current smoking, alcohol, exercise, and cholesterol. Model 3—adjusted for model 2 variables and total calories and protein intake. ^a^ Change was defined as (1992–1996 cognitive test score − 1988–1991 cognitive test score)/time between visits (years). For change, negative beta weights indicate *less* change. ^b^ For Trails B, higher scores are indicative of poorer performance.

## Data Availability

Data for this analytic study are from the publicly archived Rancho Bernardo Study and are available at https://knit.ucsd.edu/ranchobernardostudy/ (accessed 2 October 2021).

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
