# Peer review of "Egg Consumption and 4-Year Change in Cognitive Function in Older Men and Women: The Rancho Bernardo Study"

_nutrients, 2024, doi:10.3390/nu16162765_

Round 1

Reviewer 1 Report

Comments and Suggestions for Authors

Egg Consumption and 4-Year Change in Cognitive Function in Older Men and Women: The Rancho Bernardo Study

In the manuscript “Egg Consumption and 4-Year Change in Cognitive Function in Older Men and Women: The Rancho Bernardo Study”, Donna Kritz-Silverstein, et al. This article is innovative and advanced. It explores the relationship between egg consumption and changes in cognitive function of elderly men and women in the specific community, and has achieved some results. Overall, the paper discusses that egg intake is not associated with cognitive decline and suggests that it may also help maintain cognitive function over time. And found gender differences. This lays a foundation for future efforts to identify the effects of potentially modifiable factors that may prevent or delay age-related cognitive decline. Overall, the quality is acceptable, but the logic between articles needs to be strengthened and the generality of experimental data needs to be improved. In addition, the specific link between egg intake and changes in cognitive function still needs to be further studied in order to provide more accurate treatment options for future clinical diagnosis and treatment.

Below are detailed comments:

Line 21: ...egg intake was associated with less decline in category fluency... >> The format of less in the text is inconsistent with others.

Line 24-26: Results suggest that although high in dietary cholesterol, egg consumption does not have detrimental effects and may even have a role in maintenance of cognitive function. For women, there was a small beneficial effect for semantic memory. >> The tenses in the paragraphs are inconsistent. Results suggest that although high in dietary cholesterol, egg consumption does not have detrimental effects and may even have a role in maintenance of cognitive function. For women, there is a small beneficial effect for semantic memory.

Line 39-42: Although some cross-sec-tional studies report that higher dietary cholesterol intake was associated with lower cog- nitive performance [5, 6], others report either no effect [7, 8, 9], or a beneficial effect [10,11]. >> Although some cross-sec-tional studies report that higher dietary cholesterol intake is associated with lower cog- nitive performance [5, 6], others report either no effect [7, 8, 9], or a beneficial effect [10,11].

Line 96-97: Category Fluency, a test of verbal fluency, assesses semantic memory and executive function [23]. >> Category Fluency, a test of verbal fluency, assessed semantic memory and executive function [23].

Line 324-327: For instance, although both sexes showed declines in performance on Trails B and category fluency over time, at both visits, men performed better than women on Trails B and category fluency, whereas women had better performance on the MMSE. >> For instance, although both sexes have showed declines in performance on Trails B and category fluency over time, at both visits, men performed better than women on Trails B and category fluency, whereas women had better performance on the MMSE.

Line 328-329: However, egg consumption was associated with less 4-year decline only in women; a sex difference that we cannot explain. >> The tenses in the paragraphs are inconsistent. However, egg consumption was associated with less 4-year decline only in women; a sex difference that we cannot explain.

In the Table 1: The p value fonts in the table are inconsistent.

There are some questions about the content.

1. Why was four years chosen as the interval standard?

2. Is there a direct relationship between cholesterol and changes in cognitive function in older men and women? I think it would be better if the author gave a clear explanation.

Author Response

Comments 1. Line 21: ...egg intake was associated with less decline in category fluency... >> The format of less in the text is inconsistent with others.

Response 1: The word “less” is purposely written in italics for emphasis.  We have left the italics to emphasize that the result is the opposite of what would be expected if eggs (or their dietary cholesterol content) had a detrimental effect. We are willing to abide by the editor’s decision of whether or not to remove the italics (see page 1, abstract, line 23 of revised manuscript).

Comments 2:  Line 24-26: Results suggest that although high in dietary cholesterol, egg consumption does not have detrimental effects and may even have a role in maintenance of cognitive function. For women, there was a small beneficial effect for semantic memory. >> The tenses in the paragraphs are inconsistent. Results suggest that although high in dietary cholesterol, egg consumption does not have detrimental effects and may even have a role in maintenance of cognitive function. For women, there is a small beneficial effect for semantic memory.

Response 2: Thank you for pointing out this inconsistency in the text.  We have rewritten the conclusions in the abstract and this inconsistency has now been resolved (see page 1, abstract, lines 25-28).

Comments 3: Line 39-42: Although some cross-sec-tional studies report that higher dietary cholesterol intake was associated with lower cog- nitive performance [5, 6], others report either no effect [7, 8, 9], or a beneficial effect [10,11]. >> Although some cross-sec-tional studies report that higher dietary cholesterol intake is associated with lower cog- nitive performance [5, 6], others report either no effect [7, 8, 9], or a beneficial effect [10,11].

Response 3: Thank you for pointing out this inconsistency.  We have corrected it in the revised version of this manuscript (see page 1, paragraph 2, lines 43-45).

Comments 4: Line 96-97: Category Fluency, a test of verbal fluency, assesses semantic memory and executive function [23]. >> Category Fluency, a test of verbal fluency, assessed semantic memory and executive function [23].

Response 4: Thank you for noting this inconsistency in tenses.  We have made the suggested correction in the revised version of this manuscript (see page 3, paragraph 1, lines 99-100).

Comments 5: Line 324-327: For instance, although both sexes showed declines in performance on Trails B and category fluency over time, at both visits, men performed better than women on Trails B and category fluency, whereas women had better performance on the MMSE. >> For instance, although both sexes have showed declines in performance on Trails B and category fluency over time, at both visits, men performed better than women on Trails B and category fluency, whereas women had better performance on the MMSE.

Response 5: Thank you for pointing this out. We have rewritten this sentence a bit and incorporated the reviewer’s suggestion (see page 8, paragraph 4, lines 351-353).

Comments 6: Line 328-329: However, egg consumption was associated with less 4-year decline only in women; a sex difference that we cannot explain. >> The tenses in the paragraphs are inconsistent. However, egg consumption was associated with less 4-year decline only in women; a sex difference that we cannot explain.

Response 6: We thank the reviewer for pointing out the inconsistency in the tenses. In the course of addressing another reviewer’s comments, we have revised and added information to this paragraph As suggested, we have rewritten the sentence above to eliminate the inconsistency in tenses (see page 8, paragraph 4, lines 356-357).

Comments 7: In the Table 1: The p value fonts in the table are inconsistent.

Response 7: The p-value fonts in Table 1 are all in Palatino Linotype, including the p-values. However, we bolded the significant p-values.  While this may make it appear the p-values are inconsistent, we prefer it because it makes it more obvious to the reader which variables are significantly different.

There are some questions about the content.

Comments 8.1. Why was four years chosen as the interval standard?

Response 8.1: This is an older cohort.  Four years was chosen as the interval to enable us to have the largest sample size possible.

Comments 8.2. Is there a direct relationship between cholesterol and changes in cognitive function in older men and women? I think it would be better if the author gave a clear explanation.

Response 8.2: As requested, we have tried to provide a clearer explanation of the relation between cholesterol and cognitive function and report the specific results of a review and meta-analysis of seventeen studies (see page 1, paragraph 2, lines 40-42).

Reviewer 2 Report

Comments and Suggestions for Authors

I have thoroughly reviewed the manuscript titled 'Egg Consumption and 4-Year Change in Cognitive Function in Older Men and Women: The Rancho Bernardo Study,' which aims to examine the association of egg consumption with changes in cognitive function over a four-year period in a large sample of community-dwelling older adults.

The strengths of the article include a sufficient number of participants and a longitudinal study design spanning four years.

However, the manuscript could be improved in the following areas:

1. The study's ethical approval number is not provided. It is suggested to include this information in supplementary material to ensure transparency regarding study approval prior to publication.

   2. There are concerns about the accuracy of information obtained from the Willett Food Frequency Questionnaire (FFQ). Please clarify why this tool was chosen over more precise methods. Additionally, it is unclear whether general parameters such as cholesterol, glucose, HDL, and LDL levels were analyzed (if conducted, this is not described in the methodology).

3. Table 1 presents information such as education, exercise, cholesterol, diabetes, and hypertension without detailing how this information was obtained in the methodology section. Please provide a clearer description.

4. It is recommended to divide Table 1 into dietary parameters and other variables, either within the same table or as two separate tables.

5. Figure 1 lacks the quality and descriptive detail expected in a publication figure. Improvement is needed not only in its appearance but also in its description. For example, were statistical methods used to determine significant differences in the data? This should be indicated in the figure.

6. Tables from Table 2 onwards are extensive. Please consider if there is a clearer way to present this information.

7. Considering recent research on the gut-brain axis, a microbiota study conducted at the beginning and end of the experiment could strengthen the discussed findings. It is suggested to conduct such a study at the conclusion of future experiments to evaluate and discuss any differences observed.

Author Response

Comments 1: The strengths of the article include a sufficient number of participants and a longitudinal study design spanning four years.

Response 1: Thank you for recognizing the strengths of this study.

However, the manuscript could be improved in the following areas:

Comments 2: The study's ethical approval number is not provided. It is suggested to include this information in supplementary material to ensure transparency regarding study approval prior to publication.

Response 2: As requested, we have also added it directly to the text (see page 2, paragraph 4, line 79). Perhaps the reviewer missed it, but the ethical approval number for this study was originally provided in the notes following the conclusion (see page 10, paragraph 4, line 454).  We have left it in the notes.

Comments 3: There are concerns about the accuracy of information obtained from the Willett Food Frequency Questionnaire (FFQ). Please clarify why this tool was chosen over more precise methods. Additionally, it is unclear whether general parameters such as cholesterol, glucose, HDL, and LDL levels were analyzed (if conducted, this is not described in the methodology).

Response 3: The Willett Food Frequency Questionnaire has been in use for over 40 years. It has been validated against 24-hour dietary recall and biochemical indicators of nutrient intake. It was first developed in 1980 for use with the Nurses Health Study and then a revised version was released in 1985. This was available and used in the Rancho Bernardo Study beginning in 1986 for a small subsample of participants who attended the 1984-87 follow-up.  At the subsequent follow-up in 1988-91, this FFQ was given to all participants.  Other commonly used food frequency questionnaires were not available until after this baseline assessment of cognitive function.  (For example, the Block Food Frequency Questionnaire was not available until 1992.)

As requested, we have added a description of the laboratory methodology for assessment of cholesterol and glucose to the methods section (see page 3, paragraph 3, lines 131-138).

Comments 4: Table 1 presents information such as education, exercise, cholesterol, diabetes, and hypertension without detailing how this information was obtained in the methodology section. Please provide a clearer description.

Response 4: Information on the collection of education, exercise and other behavioral variables as well as medication use was provided in the original submission of our manuscript in the “Covariates” section.  As requested though, we clarified this information in this portion of the methodology section (see page 3, paragraph 3, lines 120-126) and included the fact that medical history was also obtained (including history of physician diagnosed disorders, see page 3, paragraph 3, lines 126-127). We did not add any additional information on hypertension as we had already clearly described how blood pressure was measured (see page 3, paragraph 3, lines 127-129), and how individuals were categorized as having hypertension (page 3, paragraph 4, lines 142-144) and diabetes (page 3, paragraph 4, lines 144-146). Cholesterol was used as a continuous variable in analyses, and a description of the laboratory methodology for its assessment has been added (see page 3, paragraph 3, lines 131-136).

Comments 5: It is recommended to divide Table 1 into dietary parameters and other variables, either within the same table or as two separate tables.

Response 5: As suggested, we have divided the variables in Table 1 into demographics, behaviors, health, and diet parameters (Table 1, lines 170, line 173, line 177, and line 182). We feel that it makes more sense to have these variables in a single table rather than several smaller tables.

Comments 6:  Figure 1 lacks the quality and descriptive detail expected in a publication figure. Improvement is needed not only in its appearance but also in its description. For example, were statistical methods used to determine significant differences in the data? This should be indicated in the figure.

Response 6: As requested, we have improved the quality and descriptive detail in Figure 1. Specifically, we have re-created the figure, and indicated that the statistical method used for comparisons was chi-square analysis both in the title of the table (see page 5, line 219), as well as in the text of the statistical analysis section (see page 3, paragraph 5, line 150). We have also added in more detail concerning the sample sizes for men and women at each category of egg intake (see page 5, line 220).

Comments 7: Tables from Table 2 onwards are extensive. Please consider if there is a clearer way to present this information.

Response 7: We agree that Table 2 is a large table. It was made even larger by dividing the variables under the subheadings of demographics, behaviors, health, and diet as was recommended by this reviewer for Table 1 (see comment #4 above).  We considered splitting this table by sex in two separate tables, however these tables would have likely been on two separate pages which would have made it more difficult for the reader to view and compare men and women. If the editor wishes, we are willing to omit the subheadings in this table which would make it a bit shorter in length.

We consider Table 3, to be a relatively short table.  We have chosen to leave the table as is because we feel that splitting it into tables for men and women or by cognitive function measure would make it more difficult for the reader to compare results.

Comments 8: Considering recent research on the gut-brain axis, a microbiota study conducted at the beginning and end of the experiment could strengthen the discussed findings. It is suggested to conduct such a study at the conclusion of future experiments to evaluate and discuss any differences observed.

Response 8: Thank you for your interesting suggestion and we will consider doing so in future studies. As of now though, we do not have any information on gut microbiota for this cohort.

Reviewer 3 Report

Comments and Suggestions for Authors

This study made  by Kritz-Silverstein and Bettencourton is based on the association between egg consumption and cognitive function over four years among older adults.

 The use of the Rancho Bernardo Cohort provides a robust sample of community-dwelling older adults.

Assessing cognitive function over four years adds strength to the study.

The use of various cognitive tests (MMSE, Trails B, and category fluency) provides a comprehensive assessment of cognitive function.

 The results showing a beneficial effect of egg consumption on category fluency in women but not in men suggest potential sex differences, but these findings are not very clear.

Although adjustments were made for age and education, other potential confounders that could be important in egg consumption such as overall diet quality, physical activity, and socioeconomic status were not fully explored. Is there an explanation?

The assessment of egg consumption was based on self-reported data, which can be subject to recall bias and inaccuracies. 

Some of the references included are very old. Can they be replaced with recent ones?

Also, the study does not suggest benefits for the choline, lutein, and zeaxanthin, which may play a role in cognitive health in particular, and not eggs in general. These facts should be pointed out in the text.

Author Response

Comments 1: The use of the Rancho Bernardo Cohort provides a robust sample of community-dwelling older adults.

Response 1: Thank you for recognizing the value of using this cohort!

Comments 2:  Assessing cognitive function over four years adds strength to the study.

Response 2: Thank you for recognizing this strength of the study.

Comments 3: The use of various cognitive tests (MMSE, Trails B, and category fluency) provides a comprehensive assessment of cognitive function.

Response 3: Thank you for recognizing the strength of using measures of multiple aspects of cognitive function in this study!

 Comments 4:  The results showing a beneficial effect of egg consumption on category fluency in women but not in men suggest potential sex differences, but these findings are not very clear.

Response 4: As suggested, in the discussion, we tried to clarify and point out the sex differences that we found in this study (see page 8, paragraph 4, lines 348-357).

Comments 5: Although adjustments were made for age and education, other potential confounders that could be important in egg consumption such as overall diet quality, physical activity, and socioeconomic status were not fully explored. Is there an explanation?

Response 5: We did adjust for other potential confounders in our regression analyses.  Perhaps the reviewer missed it, but as described in the statistical analysis portion of the methods section (page 4, paragraph 2, lines 161-164), as well as in the notes following Table 3 (page 7, lines 303-305), Model 1 adjusted for age and education, Model 2 adjusted for those variables plus current smoking, alcohol use, exercise and cholesterol, and Model 3 adjusted for all the variables in models 1 and 2 plus protein intake and total calories. Other models that we did not show also adjusted for diabetes and hypertension but yielded similar results.

We did not adjust for socioeconomic status because there is very little variation in this factor among the Rancho Bernardo Study participants.

Comments 6: The assessment of egg consumption was based on self-reported data, which can be subject to recall bias and inaccuracies. 

Response 6: Thank you for your comment.  We had previously noted in the limitation section that several variables used in this study were based on self-reports. However, we have now expanded this sentence to specifically mention egg consumption, and the fact that self-reports are subject to recall bias and inaccuracies (see page 9, paragraph 3, lines 394-395).

Comments 7: Some of the references included are very old. Can they be replaced with recent ones?

Response 7: We acknowledge that some of the references are very old.  However, they cannot be replaced as they are either the references used when choosing the cognitive function measures included in the 1988-91 clinic visit, or the original publications of studies comparing Rancho Bernardo Study participants to those of other large cohort studies at that time, which addresses the representativeness of this cohort. 

Comments 8. Also, the study does not suggest benefits for the choline, lutein, and zeaxanthin, which may play a role in cognitive health in particular, and not eggs in general. These facts should be pointed out in the text.

Response 8: We would like to thank the reviewer for pointing this out.  As suggested, we have pointed this out in the text (see page 9, paragraph 5, lines 419-420).

Round 2

Reviewer 1 Report

Comments and Suggestions for Authors

No more comments 

Reviewer 3 Report

Comments and Suggestions for Authors

After having read the improvements and modifications in the manuscript, I believe that this article can be published.